# Study of Rock Mass Rating (RMR) and Geological Strength Index (GSI) Correlations in Granite, Siltstone, Sandstone and Quartzite Rock Masses

**Gabor Somodi [1], Neil Bar [2], László Kovács [3], Marco Arrieta [2], Ákos Török [4] and Balázs Vásárhelyi [4,\*]**

1   Hidro-Duna Ltd., HU-2541 Lábatlan, Hungary; somodigabor@gmail.com
2   Gecko Geotechnics, Cairns, QLD 4870, Australia; neil@geckogeotech.com (N.B.); marco@geckogeotech.com (M.A.)
3   RockStudy Ltd. (Kőmérő Kft.), HU-7633 Pécs, Hungary; kovacslaszlo@komero.hu
4   Department Engineering Geolology and Geotechnics, Budapest University of Technology and Economics, HU-1111 Budapest, Hungary; torok.akos@emk.bme.hu
\*   Correspondence: vasarhelyi.balazs@emk.bme.hu

**Abstract:** A comprehensive understanding of geological, structural geological, hydrogeological and geotechnical features of the host rock are essential for the design and performance evaluation of surface and underground excavations. The Hungarian National Radioactive Waste Repository (NRWR) at Bátaapáti is constructed in a fractured granitic formation, and Telfer Gold Mine in Australia is excavated in stratified siltstones, sandstones and quartzites. This study highlights relationships between GSI chart ratings and calculated GSI values based on RMR rock mass classification data. The paper presents linear equations for estimating GSI from measured $RMR_{89}$ values. Correlations between *a* and *b* constants were analyzed for different rock types, at surface and subsurface settings.

**Keywords:** rock mass classification; RMR; GSI; granite; sandstone; quartzite

## 1. Introduction

In most aspects of deep surface and underground excavations, the focus of geological and geotechnical investigations is on understanding rock mass properties, including the mechanical properties of intact rock and discontinuities as well as the geometry and orientation of discontinuities. These conditions are evaluated according to the size of the excavation, its geometry, and available rock mass data.

Individual components of rock mass classification systems define intact rock and discontinuity characteristics as well as the orientation of discontinuities relative to an excavation and fracture frequency. Special care is required to avoid sample bias, and in most cases, discontinuity (e.g., joint) lengths can be crudely estimated by observing the discontinuity trace lengths on surface exposures [1]. The parameters of the rock mass classification systems must reflect actual ground conditions and consider their effect on the stability of the proposed excavation.

In this study, the mechanical response of the rock mass forms an essential part of understanding the behaviour of the granitic host rock at a deep underground radioactive waste disposal site (Hungary). Similarly, understanding rock mass parameters is key to assessing the stability of deep rock slope excavations at a large open pit gold mine in Australia. Scale effects, anisotropy, and confining stresses influence the rock mechanical problems, especially in geomechanics simulations [2,3]. Modelling requires geometric data regarding fracture size distribution, spatial density, and orientation. Afterwards, simulated models are available to understand features of the fractured rock body concerning hydrodynamic behaviour (e.g., connectivity, porosity, and permeability) [4] and mechanical behavior (i.e., shear strength of fractures, elastic parameters of rock mass).

Rock masses can be described using standardized descriptors and quantitative parameters associated with those descriptors, as engineers feel more confident with numbers. In this context, rock mass classification systems represent an attempt to "put numbers to geology" [5].

Various rock mass classifications have appeared since the beginning of the 20th century. Some of these include the classification system of Protodyaknov [6], of Terzaghi [7], the RQD [8], the RSR [9], the RMR [10], Q-system [11], GSI [12], RMi [13], and some recent ones such as the Rock Mass Fabric Indices (F) [14] and the Rock Mass Quality Index [15]. At this point, it should be mentioned that RMR and the Q-system have been the most widely used methods (Palmström [16]; Ranassoriya and Nikraz [17]; Fernández-Gutierrez et al. [18]). However, other classifications such as RMi and GSI have been attracting more interest in recent years.

The Rock Mass Rating (RMR) classification system was developed by Bieniawski [10] in 1973, and after several modifications, the 1989 version is the most used in rock engineering practice [19]. This classification system has been refined over several years, and its characterization method has been revised from its creation onward. The last modification of this classification system was introduced in 2014 [20].

The Geological Strength Index (GSI) system is widely used for estimating the strength reduction from an intact rock to a rock mass, introduced by Hoek [12] in 1994. It is a unique rock mass classification system used as part of the Hoek–Brown failure criterion for estimating the strength and stiffness of a rock mass [21,22]. Accurate estimation of the GSI is important for subsequent calculations [23,24]. As in other rock mass classifications, all the GSI-based equations are highly sensitive to their respective input parameters [25]. Figure 1 presents a chart for estimating GSI for jointed rock masses in the field.

**Figure 1.** Chart for determining the geological strength index (GSI) of jointed rock mass [21].

For practical engineering application, the Hoek–Brown failure criterion should only be applied when the potential for structurally controlled failures has been eliminated, as it treats the rock mass as an isotropic material.

The use and limitation of several rock mass characterization approaches are described and studied through surveyed data [26] and the results of some recent publication.

Although there are various rock mass classifications, not one of them has prevailed over another, and normally, the use of more than one classification is highly recommended,

even by their respective authors (Bieniawski [19], Palmström [16]). Since the first correlation between RMR and the Q-system, proposed by Bieniawski [27], many correlations have been proposed from tunnel and mine projects all over the world [28–30]. Moreover, when correlations are applied to the same rock lithology, better results are obtained [31–33].

The goal of this research is to prepare a comparative study to understand parameter sensitivity and differences between methods of rock mass characterization. The results help to understand relationships from mapping results from site investigations in different locations, excavation types and ground conditions, i.e., an open pit gold mine and the construction of a nuclear waste repository. This, in turn, assists in understanding potential gaps and sources of error when correlations are used, e.g., estimated from drill core data.

## 2. Research Methodology and Research Materials

*Determination of GSI and RMR$_{89}$*

GSI can be estimated for an exposed rock slope or tunnel face using the standard chart and field observations of rock mass blockiness and surface condition of the discontinuity (see Figure 1). Other charts are available for different rock masses (e.g., flysch, foliated, etc.).

Several independent quantitative theories were developed for the estimation of GSI values from drill core data. This has been used in cases when rock exposures, tunnel faces or slopes were not yet available (e.g., initial feasibility studies prior to excavation). Different rock mass classification systems have been utilized in the approach to correlate drill core data to GSI, and include the following:

- Q-system-based calculation method (e.g., [34,35])
- RMi-based calculation method (e.g., [36,37])
- RMR-based calculation method (e.g., [21,35])

Several authors [26,38–40] have theoretically shown that significant differences in GSI estimation result from applying different calculation methods for different rock types.

This paper focuses on calculated estimations of GSI from Rock Mass Rating (RMR) values. Recently, several researchers have investigated the relationship between Bieniawski's [19] Rock Mass Rating (RMR$_{89}$) and the GSI. Firstly, [21,33] suggested the following calculation:

$$GSI = RMR_{89} - 5 \text{ for } RMR_{89} > 23 \tag{1}$$

In this equation, RMR$_{89}$ can be calculated from the following parameters:

$$RMR_{89} = R1 + R2 + R3 + R4 + R5 \tag{2}$$

where:

- R1—uniaxial compressive strength (0–15),
- R2—rock quality designation, RQD (3−20),
- R3—average joint space (5–20),
- R4—joint wall conditions (0−30), and
- R5—water. In the original definition, R5 must be defined as dry (i.e., 15) for assessing drill core.

Based on further studies, Hoek et al. [35] suggested the following simple formula for GSI calculation:

$$GSI = 0.5 \text{ RQD} + 1.5 \text{ J}_c \tag{3}$$

where J$_c$ is the joint wall conditions, according to the definition of Bieniawski [19] and it is equal to the R4 value of the previous equation, thus the rate of it is between 0 and 30.

Similarly, Hoek et al. [22,33] and several authors have recommended estimating the GSI value from RMR$_{89}$ using a linear equation in the form of

$$GSI = a \text{ RMR}_{89} + b \tag{4}$$

where a and b are different constants, depending on the rock type or geographical location. Table 1 summarizes various constants that have been calculated in previous works.

**Table 1.** Equations of the existing correlations between $RMR_{89}$ and GSI.

|   | a | b | $R^2$ | Rock Type | Ref. |
|---|------|---------|-------|-----------|------|
| 1 | 1.00 | $-5$ | | various projects | Hoek et al. [22] |
| 2 | 0.42 | 23.08 | 0.44 | schist and sedimentary rocks | Cosar [41] |
| 3 | 0.739 | 12.097 | 0.759 | sandstone | Irvani et al. [42] |
| 4 | 0.7394 | $-4.3349$ | 0.57 | metamorphic | Singh and Tamrakar [43] |
| 5 | 0.9932 | $-4.913$ | 0.84 | gabbro, ultrabasic | Ali et al. [44] |
| 6 | 1.2092 | $-18.6143$ | | various projects | Zhang et al. [45] |
| 7 | 1.265 | $-21.49$ | | various types of rocks | Siddique and Khan [46] |

In case of poor and very poor rock mass (i.e., RMR < 30) Osgoui and Ünal [47] suggested an exponential relationship for calculating the GSI value from $RMR_{89}$:

$$GSI = 6e^{0.05RMR_{89}} \tag{5}$$

The suggested equations (using the constants presented in Table 1) are summarized in graphic form (Figure 2). Reviewing the results, it is evident that the equation of Osgoui and Ünal [47] is the only one focusing on $RMR_{89}$ below 30. The results also suggest that equations by Cosar [41] and Singh and Tamarkar [43] underestimate GSI for $RMR_{89} > 30$ when compared to the others. The remaining five formulae converge with similar results between $RMR_{89}$ values of 60 and 70, while drifting apart above and below. As illustrated in Figure 2, the highest variability between equations occurs when $RMR_{89} < 30$, and when $RMR_{89} > 80$. These represent rock masses at opposite ends of the spectrum, those of very poor, and very good quality.

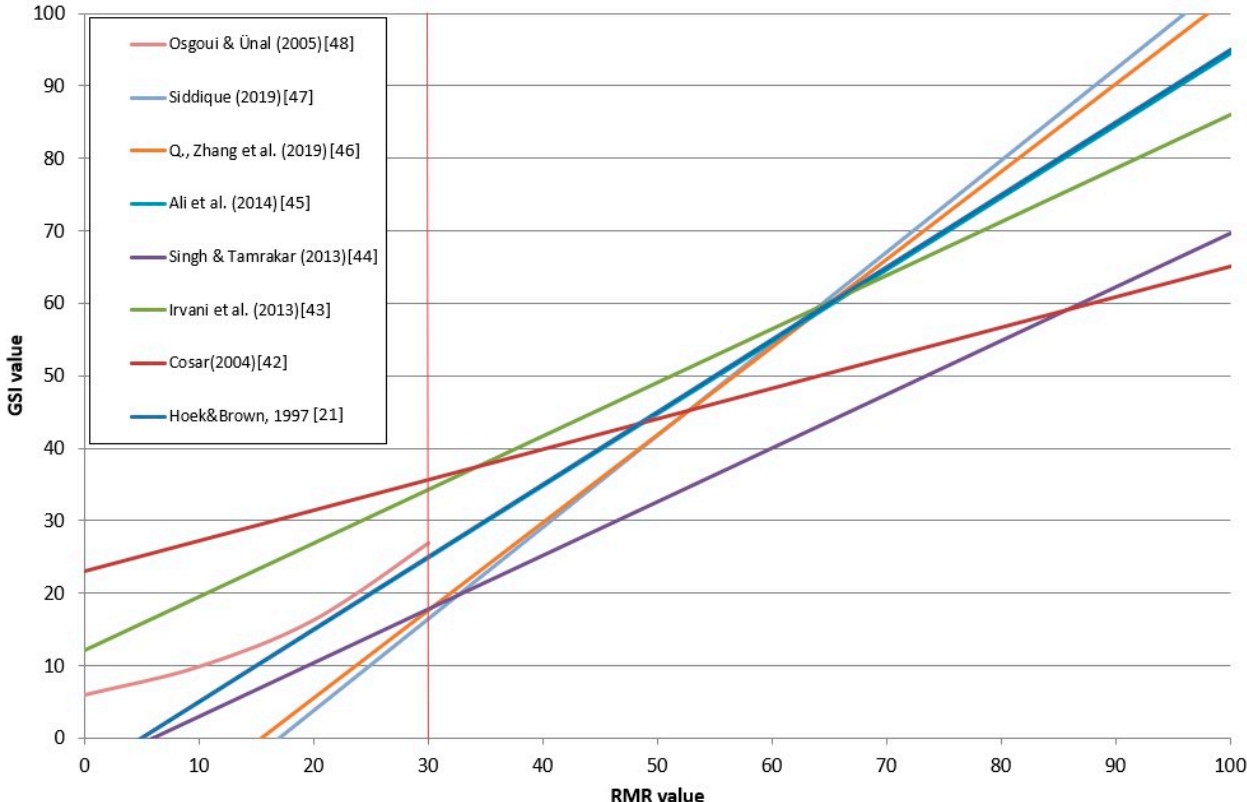

**Figure 2.** Existing correlations between $RMR_{89}$ and GSI listed in Table 1.

## 3. Geological Setting and Database

### 3.1. Geological Setting and Data of Hungarian Nuclear Waste Repository

The research areas comprise granitic rock masses near Bátaapáti in Hungary and anisotropic siltstones, sandstones, and quartzite in Western Australia.

The fractured granitic body forms the host rock of the Hungarian National Radioactive Waste Repository (NRWR). Until now, a more than 6 km long tunnel network has been constructed, and all the tunnel faces have been documented [4,48]. From a geological aspect, three main rock types can be distinguished in the carboniferous granite formation: monzogranite, monzonite, and hybrid rocks [3,49]. This granitic body is transected by Cretaceous trachyandesite dykes with NE–SW strike and randomly distributed aplitic veins also [50]. The four repository chambers were excavated in monzogranite with aplitic veins and scarce monzonite enclaves.

The results of previous field observations and models suggested that the granite formation is hydraulically strongly compartmented, dividing the underground flow system into several blocks of limited hydraulic connection. Based on field observations, it was distinguished as less transmissive blocks bordered by more transmissive zones. The repository for low and medium level nuclear waste disposal is placed in a less transmissive hydraulic compartment [51,52]. Based on measured geometric data (spatial position, length, orientation, and aperture), fracture networks are simulated to study connectivity relations and for computing the fractured porosity and permeability at different scales. The results prove the scale-invariant geometry of the fracture system [4,53]. Geotechnical data from drill core logging and tunnel face documentation show a clear connection between rock mass classification and rock type [26,49,50].

The database consists of two types of $RMR_{89}$ and GSI values: field values, which were estimated directly from slope or tunnel faces, and values back-calculated from other methods (e.g., GSI from $RMR_{89}$). The database covers the ratings from repository chambers and other tunnels with smaller cross-sections (Figure 3).

### 3.2. Geological Setting and Data of Telfer Gold Mine

A large gold deposit occurs within Proterozoic stratigraphy at the Telfer Gold Mine in the Great Sandy Desert of Western Australia. Rock types include calcareous and argillaceous siltstones, sandstones and quartzites. The geological structure is complex and is the primary reason behind the mineralization [54].

The properties of intact rock mass and rock mass shear strength, and to a lesser extent, bedding shear strength, vary with the degree of weathering and the type of alteration (clay or silica enrichment). Planar sliding along adversely oriented bedding planes within siltstone, sandstone and quartzite are the most common causes of slope instability.

Similarly to the Hungarian granitic rock masses, two types of data exist for Telfer: field values of GSI and $RMR_{89}$, estimated on-site based on observations from 12–24 m high slope faces (Figure 4), and calculated GSI values from drill core estimations of Q and $RMR_{89}$ [55].

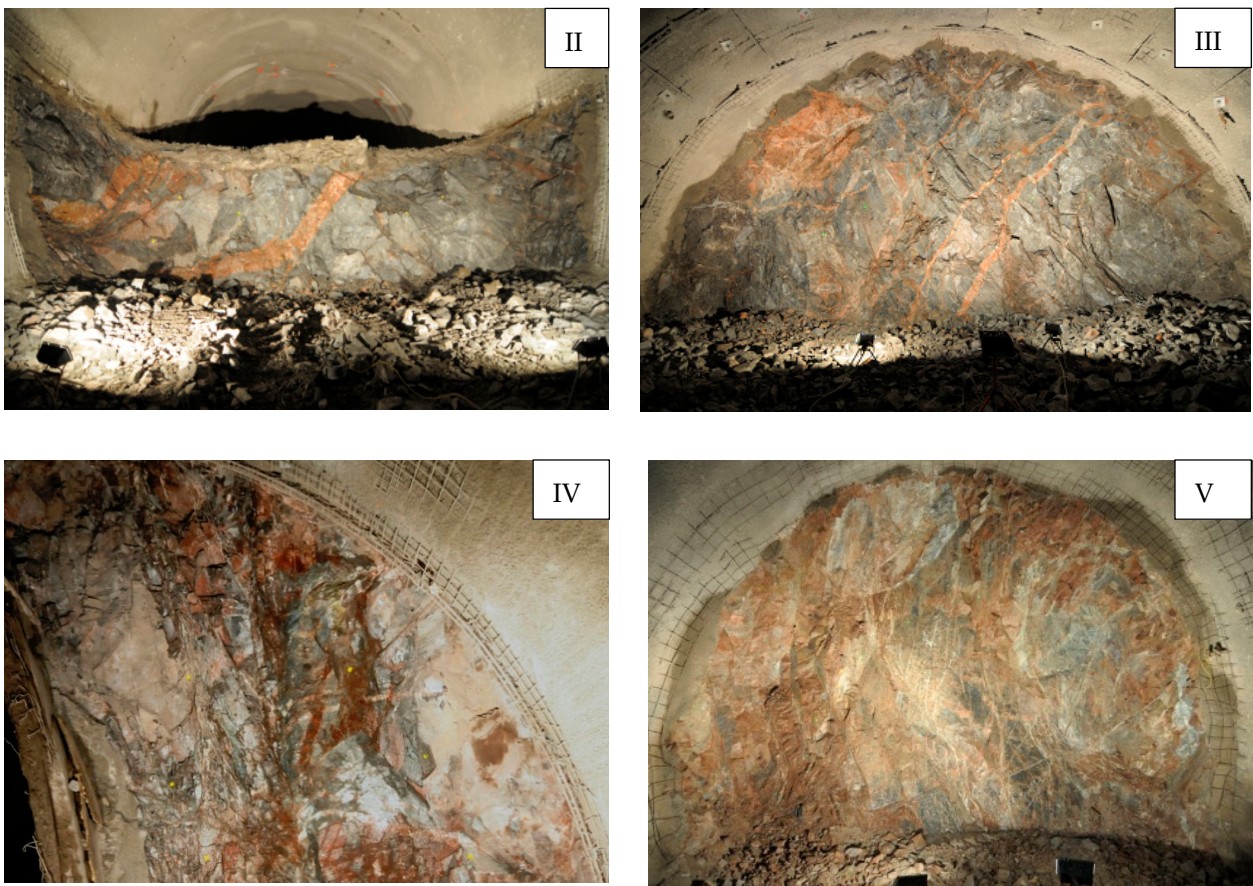

**Figure 3.** Examples of rock mass classes in various tunnel sections: Chamber bench ~ 60 m$^2$ (class **II**), Chamber top heading ~ 60 m$^2$ (class **III**), Chamber half top heading 30 m$^2$ (class **IV**), Research tunnel 33 m$^2$ (class **V**) (granitic rocks, Bátaapáti, Hungary).

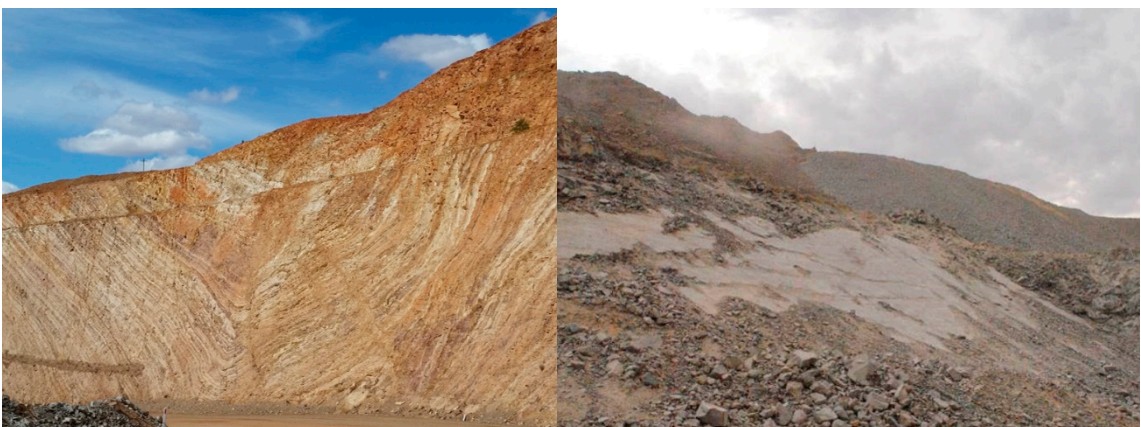

**Figure 4.** (**Left**): near-laminated weathered siltstone with siltstone-siltstone bedding planes (Telfer Gold mine, Australia). (**Right**): planar failure in fresh quartzite with quartzite-quartzite bedding planes (Telfer Gold Mine, Australia). Both have bedding scale anisotropy [28].

## 4. Results and Discussion

### 4.1. Comparing the Calculated GSI Values to the Chart Values

Using the entire database, the two RMR-based different equations of GSI determination (Equations (1)–(5)) and GSI field values were analyzed, based on the observations

during the excavation of the Bátaapáti radioactive waste repository. Using the two RMR-based equations, Equations (1) and (3), different correlations were found (Figures 5 and 6). Using Equation (1), the obtained values are always higher than the chart values ($GSI_{chart}$). There is a linear relationship between the two values:

$$GSI_{chart} = 0.793\ GSI_{calc(1)} + 5.974\ (R^2 = 0.736) \tag{6}$$

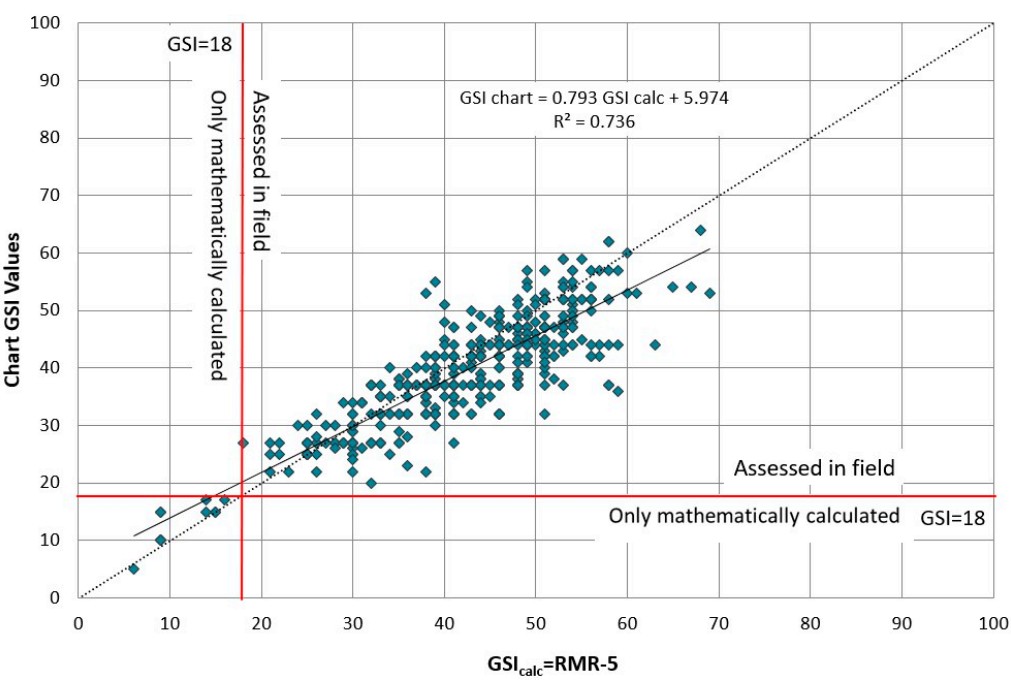

**Figure 5.** Correlations between Chart and calculated GSI values using Equation (1).

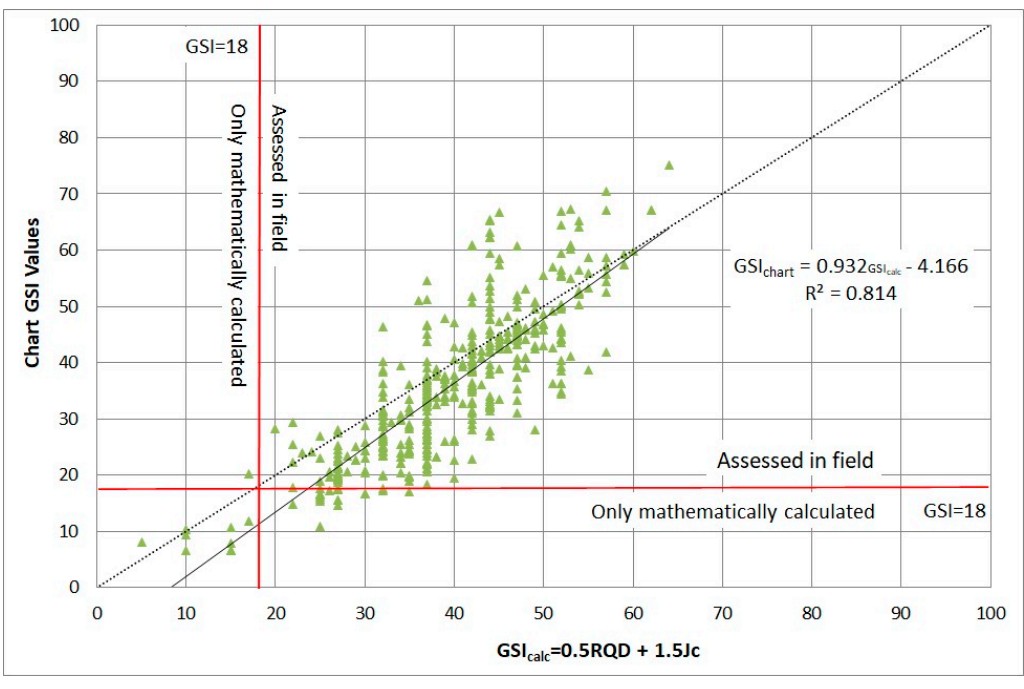

**Figure 6.** Correlations between Chart and calculated GSI values using Equation (3).

Using Equation (3), the calculated points are nearly equal to the chart points:

$$GSI_{chart} = 0.932\ GSI_{calc(2)} - 4.166\ (R^2 = 0.814) \tag{7}$$

### 4.2. Comparing the RMR Value to the GSI Value

The chart Geological Strength Index is plotted as a function of $RMR_{89}$ values in Figure 7. According to the calculated results, the linear regression is shown in Table 2.

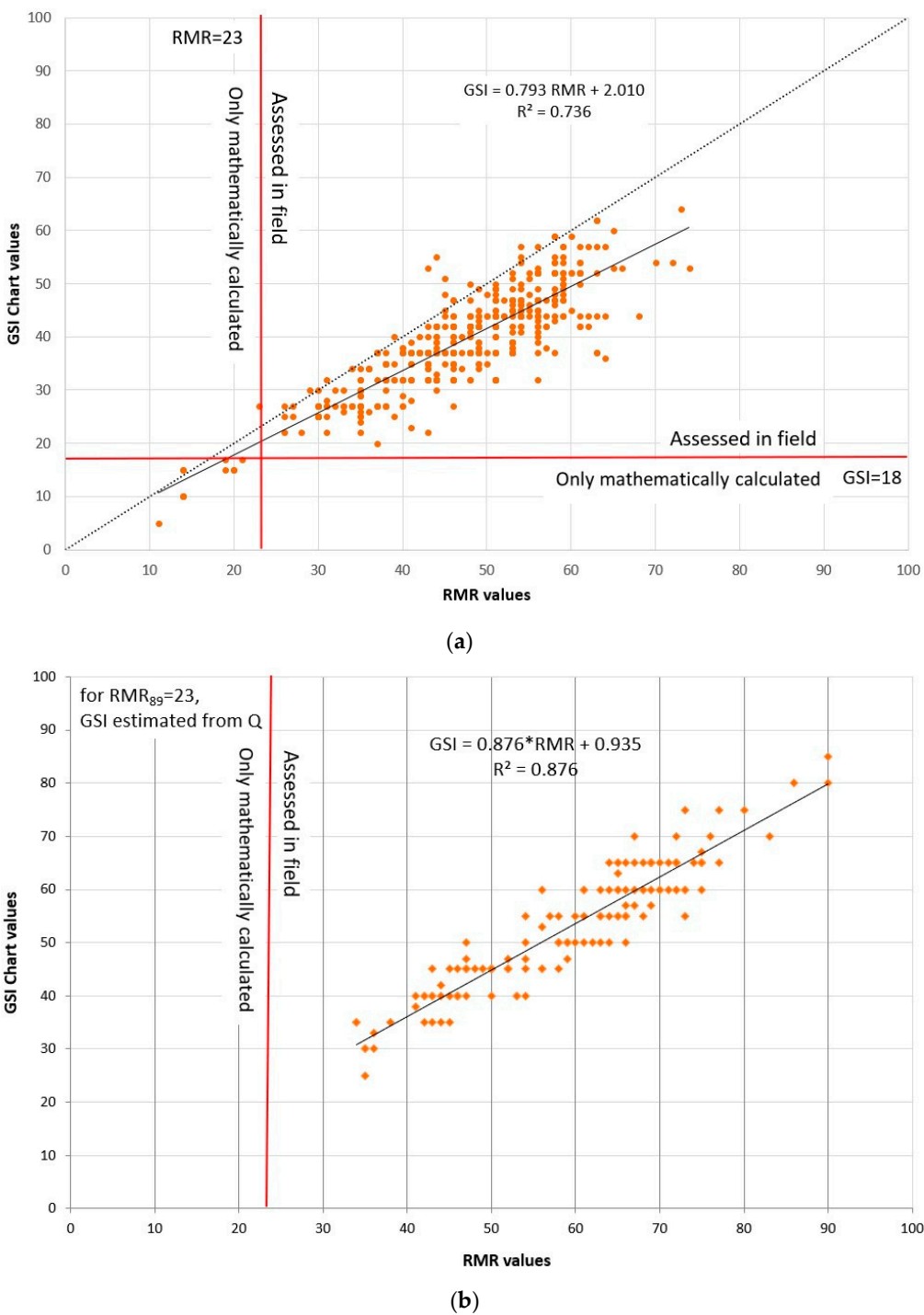

**Figure 7.** Correlation between Chart GSI and RMR values in Bátaapáti project (**a**), and in Telfer project (**b**).

**Table 2.** Equations of the existing correlations between $RMR_{89}$ and GSI in the studied projects.

| a | b | $R^2$ | Rock Type | Ref. |
|---|---|---|---|---|
| 0.793 | 2.010 | 0.736 | granitic rocks | Bátaapáti |
| 0.876 | 0.935 | 0.876 | siltstones, sandstones and quartzites | Telfer |

The studied correlations show distinct connections between the two rock mass characterization methods. Good correlations between a and b constants from Equation (3) were identified in different projects.

The use of rock-type-based equations and constants from Table 1 did not identify a clear or discernible correlation between $RMR_{89}$ and GSI. However, further examination of the relationship between 'a' and 'b' constants indicate a strong correlation for individual cases. The connection between a and b constants obtained from Equation (4) are illustrated in Figure 8.

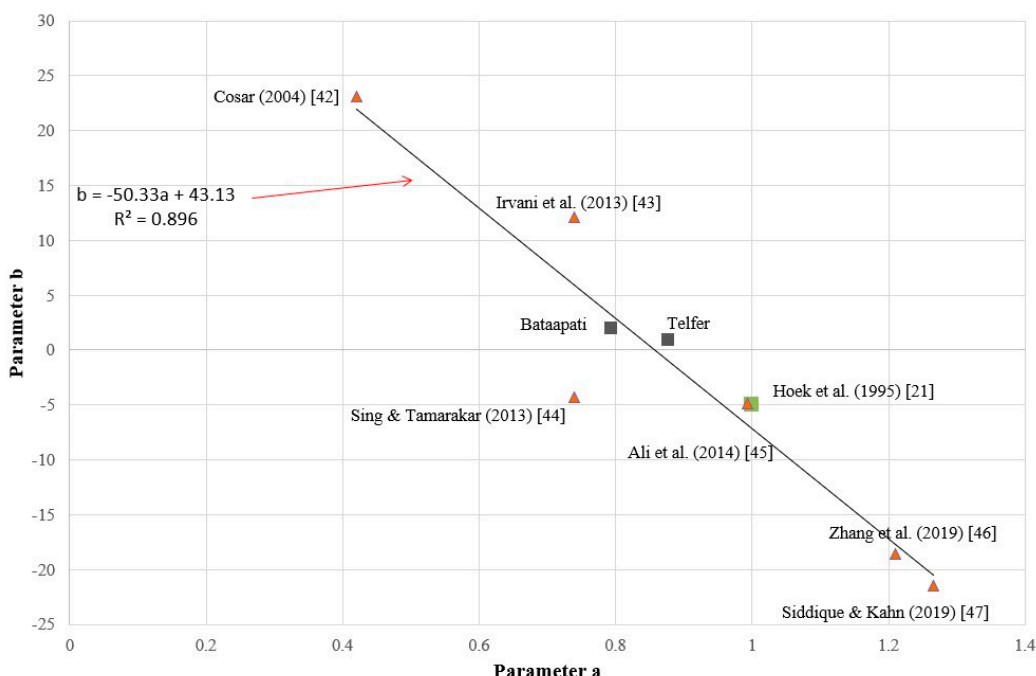

**Figure 8.** The correlation between different a and b constants regarding different correlations between GSI-$RMR_{89}$ values.

The calculated values from Hungary (Bátaapáti) and Australia (Telfer) from this paper are also shown in Figure 8. Constants a and b from Equation (4) can be correlated by:

$$b = -50.13a + 43.13 \ (R^2 = 0.896) \tag{8}$$

This value is considered as site- and rock-type dependent.

In Figure 8, the various relationships appear to follow a linear relationship with the exception of published data by Singh and Tamarkar [43]. When these data are omitted, an even higher correlation is attained:

$$B = -53.77a + 47 \ (R^2 = 0.977) \tag{9}$$

Using this latter Equation (9), a = 1, while b = −6.29. These values are commensurate to the values suggested by Hoek and Brown [33].

## 5. Conclusions

This study describes the relationship between Rock Mass Rating (RMR) and Geological Strength Index (GSI). Quantitative assessments of the correlations have been conducted systematically based on the available study data from Bátaapáti and Telfer on the correlations of primary classified indices of RMR and GSI.

The following simplified quantitative correlations between $RMR_{89}$ and GSI are proposed using large data sets:

$$GSI = 0.793\ RMR_{89} + 2.001\ (R^2 = 0.736,\ Bátaapati) \tag{10}$$

$$GSI = 0.876\ RMR_{89} + 0.935\ (R^2 = 0.876,\ Telfer) \tag{11}$$

These two proposed quantitative correlations between $RMR_{89}$ and GSI are applied to evaluate the surrounding rock mass of the Bátaapati site (granitic rock mass in Hungary) and Telfer Gold Mine (siltstones, sandstones and quartzites of Western Australia). Whilst they are likely applicable elsewhere in similar ground conditions, readers are strongly encouraged to validate these and other correlations at their local site prior to use.

The validated results demonstrate that the proposed simplified quantitative correlation reflects the observed relationship between $RMR_{89}$ and GSI.

To provide the input data for empirical design of excavations in tunnels or mining projects, it is necessary to determine the geological conditions in the study areas and carry out rock mass classifications to predict ground behavior. The obtained results can be also applied in other research fields such as geomechanics simulation, fracture network characterization and hydrological modelling. Based on these results, it can be concluded that an extensive data set of underground or surface settings provides a sound basis for rock mass classification of various lithologies.

**Author Contributions:** G.S. Writing—original draft preparation, L.K. and Á.T.; Formal and data collection analysis, N.B. and M.A.; Reviewing, methodology and data collection, B.V.; Conceptualization, investigation and methodology, editing. All authors have read and agreed to the published version of the manuscript.

**Funding:** This research was funded by National Research Development and Innovation Office of Hungary (Grant No. TKP2020 BME-IKA-VIZ and NKFIH 124366, NKFIH 124508) and the Hungarian-French Scientific Research Grant (No. 2018-2.1.13-TE T-FR-2018-00012).

**Institutional Review Board Statement:** Not applicable.

**Informed Consent Statement:** This paper has been published with the permission of Public Limited Company for Radioactive Waste Management (PURAM).

**Data Availability Statement:** 3rd Party Data.

**Conflicts of Interest:** The authors declare no conflict of interest.

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
