# Peer review of "Study of Rock Mass Rating (RMR) and Geological Strength Index (GSI) Correlations in Granite, Siltstone, Sandstone and Quartzite Rock Masses"

_applsci, doi:10.3390/app11083351_

Round 1
Reviewer 1 Report
Citing order needs to be relative (starting from 1) during the text. This improves the reading experience and eases reference finding. Please also improve the legibility of all graphs (font size).
The guidelines for utilization of GSI are clear in stating that is preferable to use a range rather than a precise value for describing the rockmass. How is this achieved/explained and/or addressed in the research?
Is the joint condition, jC = jR × jL/jA?. If so, this needs to be explained. Why are you mentioning this correlation and how is it used in your research afterwards?
I suggest adding sub-chapters 3.1 and 3.2 for explaining the geological setting of both sites.
How is the hydraulic transmissivity of blocks visually assessed? What is the confidence of these observations? is there any testing/lab data to support these observations? How is this data used in your research? correlation to rockmass quality? joint alteration number?
When mapping a fractured rockmass there are usually 2 or more joint sets (the same can be seen in the attached pictures) How many joint sets are in each area? Is it possible to provide pole plots and/or rose diagrams?
In the same line, estimation of parameters R3 and R4 within RMR is usually done considering the "worst case scenario" for a given joint set, i.e. the one that is affecting stability the most. Is there any verification of how RMR was assessed within your database? This might highly influence the correlation that you are attempting to find afterwards. How a range of plus/minus 5 point of GSI will affect this correlation?
I suggest a better explanation of each database. For example "calculated values determined by several other concepts of RMR89 and GSI methods using the same field observations" seems too vague. What concepts?
In Figure 4, there is no purpose in showing and using values under 18 for GSI=RMR-5, since this has no sense by definition of the correlation by Hoek.
Table 1 and all the text linked to it should be in the introduction, since this is part of the literature review and not of your results.
Figure 7 is very difficult to read. The value presented in the trend line is different to both presented in equations 8 and 9.
How the accuracy of your proposed equation compares to the other formulas proposed before? Can you describe a quantitative improvement over the stat of the art techniques for your particular case?
It will be a good contribution if you can suggest a methodology to find such correlations to "any" rock excavated environment, so authors and industry might find a useful tool as a result of your investigation.
Author Response
First of all, we would like to thank the reviewer for her/his work. Our answers
Citing order needs to be relative (starting from 1) during the text. This improves the reading experience and eases reference finding. Please also improve the legibility of all graphs (font size).
Thank you for your comment. In the revised paper it was modified accordingly.
The guidelines for utilization of GSI are clear in stating that is preferable to use a range rather than a precise value for describing the rockmass. How is this achieved/explained and/or addressed in the research?
The field data is always determine as a range, but for presentation we use the averages of determined ranges of field values. This has been remarked in the corrected paper.
Is the joint condition, jC = jR × jL/jA?. If so, this needs to be explained. Why are you mentioning this correlation and how is it used in your research afterwards?
It wac determined according to BIeniawski (1989) and as it is suggested by Hoek et al. (2013) GSI can be calculated using this value. We use this equation throughout the whole paper.
I suggest adding sub-chapters 3.1 and 3.2 for explaining the geological setting of both sites.
Thank you for your suggestion: new sub-chapters were introduced.
How is the hydraulic transmissivity of blocks visually assessed? What is the confidence of these observations? is there any testing/lab data to support these observations? How is this data used in your research? correlation to rockmass quality? joint alteration number?
It is corrected. The original sentence can seuggest falsely that the hydraulic transmissivity of blocks visually assessed. The rock mass data gave some field observation to create a good hidrogeologic concept – see in Benedek, 2009, M. Tóth 2010, and 2018 – but of course the main field data was the hydraulic transmissivity measurements in boreholes. Field mapping was needed only to create a useful model.
When mapping a fractured rockmass there are usually 2 or more joint sets (the same can be seen in the attached pictures) How many joint sets are in each area? Is it possible to provide pole plots and/or rose diagrams?
This is not really relevant to the topic of the paper. The data is there to show this, but firstly, there are over 100 cases per site. Structure orientation patterns vary with spatial location. As such, a single pole plot across the site would ‘average out into main sets which will not represent each individual location. The number of structure sets are counted visually in the field for RMR/GSI as appropriate. The photographs are merely shown as ‘typical cases’ for the reader to have an idea of the ground conditions, not looking at specifics of an individual case.
In the same line, estimation of parameters R3 and R4 within RMR is usually done considering the "worst case scenario" for a given joint set, i.e. the one that is affecting stability the most. Is there any verification of how RMR was assessed within your database? This might highly influence the correlation that you are attempting to find afterwards. How a range of plus/minus 5 point of GSI will affect this correlation?
RMR was estimated as per the method suggestions which you mention below. Data was sourced by a geotechnical engineer and reviewed by a senior/principal. Purpose of the paper is to demonstrate that the relationship varies depending on the ground conditions. Yes a plus/minus of 5 has an impact; however, this impact is far less than that observed for different ground conditions. The focus on the paper is not ‘user error”. “Perception of GSI” using charts can be an entirely separate paper – we have some data from a mining company where 20 engineers were asked to assess exactly the same slopes – variability in results up to 10% (largely due to junior, inexperienced staff). As such we used a peer review process here.
I suggest a better explanation of each database. For example "calculated values determined by several other concepts of RMR89 and GSI methods using the same field observations" seems too vague. What concepts?
Statement improved – thank you for highlighting
In Figure 4, there is no purpose in showing and using values under 18 for GSI=RMR-5, since this has no sense by definition of the correlation by Hoek.
The figures were corrected, GSI =18 and RMR=23 have been marked in the revised figures. (Fig 5. and Fig 6.)
Table 1 and all the text linked to it should be in the introduction, since this is part of the literature review and not of your results.
Thank you for your suggestion it is revised.
Figure 7 is very difficult to read. The value presented in the trend line is different to both presented in equations 8 and 9.
It is revised and corrected, thank you for the comment.
How the accuracy of your proposed equation compares to the other formulas proposed before? Can you describe a quantitative improvement over the stat of the art techniques for your particular case?
New equation was suggested based on hundreds of GSI and RMR measurements, and the large data set allowed us to calculate the values with higher confidence. It is clean that the data scatter is large due to the lithological and structural differences.
It will be a good contribution if you can suggest a methodology to find such correlations to "any" rock excavated environment, so authors and industry might find a useful tool as a result of your investigation.
We are trying to demonstrate the opposite – i.e. that you need to find a relationship for your own ground conditions, and/or validate existing ones to determine that they are suitable rather than blindly applying a correlation without understanding its limitations.
The equations given in this paper works well for the studied sites and rock types.
Reviewer 2 Report
Dear authors, you have written an interesting paper, but the following should be corrected in it:
Line 20: The literal expression of the equation should not be in the Abstract. There should be a full stop after the word "values".
Line 25: A more detailed literature review is missing in the chapter Introduction. It is necessary to put other papers from Professor Hoek then Professor Marinos, and others. The references are not listed in order of appearance in the text. That needs to be corrected, necessarily!
Line 58, 59: This statement is not true! The Geological Strength Index (GSI) was introduced by Hoek in Hoek, E. Strength of rock and rock masses, ISRM News Journal, 1994. 2(2), 4-16. That paper should be cited in the text and in the References.
Line 72: Figure 1 must be behind the text when it first appears. It appears on line 77, and the current place within the introduction chapter is not appropriate.
Line: 75: The chapter on research methodology and research materials that preceded the writing of the paper is missing. It needs to be formed. Chapters 2 and 3 may be adapted for this purpose. So, the title of this chapter “2. Determination of GSI” should become the title of the second line within the new chapter.
Line 106: There must be a second-order title within the chapter on methodologies and research materials.
Line 153: Add words "and discussion"
Line 169: It is not enough to list only R2 as a measure of the success of the equation! Why is not calculated Adj. R2? It is a more appropriate measure of success, and there are others. Authors must at least try to explain why they have not established a nonlinear equation. What would be her R2 and Adj. R2. Also, does exist a verification process for this regression model?
Line 179: It is not appropriate that the word "(see Figure 5 also)" being in the title of Table 1. Also, in Table 1 in column 6, the names of the authors should be written next to the reference number.
Line 207: What does "higher correlation" mean? It is necessary to specify the value ranges for R2 which mean "higher correlation"! It is necessary to cite the literature in which the meaning of each range of R2 values is described.
Line 211: In Figure 7, the equation must contain "b" instead of "y" and "a" instead of "x"! The hypen to minus in figure needs to be changed. Decimal point, (,) and (.) are mixed, so form unification is necessary.
Line 225 – 227: To which part of the paper does this claim refer?
Line 242: This list must be formatted according to the instructions for authors!
I wish you all the best!
Author Response
Line 20: The literal expression of the equation should not be in the Abstract. There should be a full stop after the word "values".
Thank you for your comment it is revised
Line 25: A more detailed literature review is missing in the chapter Introduction. It is necessary to put other papers from Professor Hoek then Professor Marinos, and others. The references are not listed in order of appearance in the text. That needs to be corrected, necessarily!
Thank you for your remark, additional papers are cited.
Line 58, 59: This statement is not true! The Geological Strength Index (GSI) was introduced by Hoek in Hoek, E. Strength of rock and rock masses, ISRM News Journal, 1994. 2(2), 4-16. That paper should be cited in the text and in the References.
The authors modified the paper in accordance with this comment.
Line 72: Figure 1 must be behind the text when it first appears. It appears on line 77, and the current place within the introduction chapter is not appropriate.
Its position was corrected, thank you for the remark.
Line: 75: The chapter on research methodology and research materials that preceded the writing of the paper is missing. It needs to be formed. Chapters 2 and 3 may be adapted for this purpose. So, the title of this chapter “2. Determination of GSI” should become the title of the second line within the new chapter.
Line 106: There must be a second-order title within the chapter on methodologies and research materials.
The authors modified the paper in accordance with these two comments, too.
Line 153: Add words "and discussion"
It is revised accordingly.
Line 169: It is not enough to list only R2 as a measure of the success of the equation! Why is not calculated Adj. R2? It is a more appropriate measure of success, and there are others. Authors must at least try to explain why they have not established a nonlinear equation. What would be her R2 and Adj. R2. Also, does exist a verification process for this regression model?
The suggested equations are based on large set of data, it also means that R2 in its own provides a good overview of the data scatter and variability. As it is proposed by Ratner (2009) R2 also provides a good insight to data correlation and adj R2 is overused.
Ratner, B (2009): The correlation coefficient: Its values range between + 1 / − 1, or do they? Journal of Targeting, Measurement and Analysis for Marketing Vol. 17, 2, 139–142
Line 179: It is not appropriate that the word "(see Figure 5 also)" being in the title of Table 1. Also, in Table 1 in column 6, the names of the authors should be written next to the reference number.
Thank you for your suggestions all of these have been considered and the relevant parts are revised.
Line 207: What does "higher correlation" mean? It is necessary to specify the value ranges for R2 which mean "higher correlation"! It is necessary to cite the literature in which the meaning of each range of R2 values is described.
When R2 is higher than 0.7 it is considered a good correlation (Ratner, 2009)
Line 211: In Figure 7, the equation must contain "b" instead of "y" and "a" instead of "x"! The hypen to minus in figure needs to be changed. Decimal point, (,) and (.) are mixed, so form unification is necessary.
Thank you for your remarks – corrections are made.
Line 225 – 227: To which part of the paper does this claim refer?
This statement refers to the studied lithologies and the equations are best applied for these types of sites.
Line 242: This list must be formatted according to the instructions for authors!
It is revised.
Reviewer 3 Report
The manuscript presents the correlation between two rock mass classification, the Rock Mass Rating (RMR) and Geological Strength Index (GSI) in various rock masses. The idea of correlating various rock mass classifications but it shows some interesting findings.
Nonetheless, some points must be corrected or further information must be included.
First of all, it must be said that, according to the journal’s template, references must be ordered as they appear in the paper, not in other order. Therefore, the first cite we find in the paper must be [1] and not [27] (line 37). Consequently, all the references must be reordered.
Secondly, I will extend the introduction of the rock mass classification. I see a gap between paragraph in lines 49-52 and subsequent paragraphs.
Once the necessity of the rock mass classification has been introduced in the paper, it should be mentioned that, various rock mass classifications appeared since the beginning of the 20th century. Some of those classifications were the system of Protodyaknov (1907), of Terzaghi (1946), the RQD (Deere et al. 1967), the RSR (Wickham et al. 1972), the RMR (Bieniawski, 1973), Q system (Barton et al. 1974), GSI (Hoek et al. 1995), RMi (Palmström, 1996), and some recent ones like the Rock Mass Fabric Indices (F) (Tzamos and Sofianos, 2007) and the Rock Mass Quality Index (Aydan et al. 2014). At this points, it should be mentioned that RMR and the Q system are the most widely employed methods (Palmströn, 2009; Ranassoriya and Nikraz 2009; Fernández-Gutierrez et al. 2017). However, other classifications like RMi and GSI are attracting more and more interest. Then, the RMR and the GSI can be described, continuing with existing text.
In line 53, I will remark that the RMR was developed by Bieniawski in 1973 [and its reference), to highlight that later, in 1989, the most used version emerged.
Similarly, in line 59 it is beneficial to introduce that Hoek et al. introduced the GSI in 1995. Thus, the time gap between both classifications can be better observed.
Line 62-63. I think that this idea, that the GSI-based equations are sensitive to their respective input parameters is a universal idea that is verified for all the rock mass classifications. Therefore, it could be added, “As in other rock mass classifications, all the GSI-based equations are highly...”
After line 64 or after line 66, I will include a very important idea: there are researchers in the literature about correlating rock mass classifications between them. Therefore, it could be included something like: “Although there are various rock mass classifications, no one of them have imposed above the other and normally, the employment of more than one classification is highly recommended, even by the authors of them (Bieniawski, 1989, Palmströng, 2009). Since the first correlation between RMR and the Q index, proposed by Bieniawski (1976), many correlations have been proposed from tunnel and mine projects all over the world (Rutledge and Preston, 1978; Moreno Tallón 1982, Sayeed and Khanna 2015). Moreover, when correlations are applied to the same rock lithology, better results are obtained (Castro-Fresno et al. 2010, Fernández-Gutierrez et al. 2017; Campos et al. 2020).
Line 77. Fig 1 cannot be presented before it is commented in the text.
Last but not least, for section 3 and 4, a better organization is needed. Available data must be better explained. I am really confused about the data that were analyzed. Data were obtained by observing in field. Then, what is GSIcalc? In my opinion, if authors have RMR and GSI data from different tunnel/mines, they can show the scatter plots and obtained correlations are presented. It is not necessary to calculate GSI from other proposed equations. It seems that authors try to verify others’ equations instead of developing their own correlations. A deep review of the procedures iof section 4 is needed. Moreover, when presenting equations like in Figure 6, each of the references must be linked, i.e. its reference in square brackets must be given [xx]. Similarly, for Figure 7.
Aydan et al. (2014). A new rock mass quality rating system: Rock Mass Quality Rating (RMQR) and its applications to the estimation of geomechanical characteristics of rock masses. Rock Mechanics and Rock Engineering, 47(4): 1255-1276
REFERENCES
Barton et al. (1974). Engineering classification of rock masses for the design of rock support. Rock Mechanics. 6(4): 189-236, doi: 10.1007/BF01239496.
Bieniawski (1973). Engineering classification of jointed rock masses. South African Institute of Civil Engineers, 15(12): 333-343.
Bieniawski (1976). Rock mass classification in rock engineering. In Bieniawski, Z. T. (Ed.), Proceedings of the Symposium on Exploration for Rock Engineering (97-106). Johannesburg: A. A. Balkema.
Campos et al. (2020). New GSI correlations with different RMR adjustments for an eastern mine of the Quadrilatero Ferrífero. Journal of South American Earth Sciences, 102, 102647, doi: 10.1016/j.jsames.2020.102647
Castro-Fresno (2010). Correlation between Bieniawski’s RMR and Barton’s Q index in low-quality soils. Revista de la Construcción, 9(1): 107-119.
Deere et al. (1967). Design of surface and near-surface construction in rock. In Fairhurst (Ed.), Failure and breakage of rock, proceedings 8th US symposium on rock mechanics (pp. 237-302). New York: Society of Mining Engineers, AIME.
Fernández-Gutierrez et al. (2017). Correlation between Bieniawski’s RMR index and Barton’s Q index in fine-grained sedimentary rock formations. Informes de la Construcción, 69(547), e205, doi: 10.3989/id54459
Hoek et al. (1995). Support of Underground Excavations in Hard Rock. Rotterdam: A. A. Balkema.
Moreno Tallon (1982. Comparison and application of geomechanics classification schemes in tunnel construction. In Tunnelling 82, 3rd International Symposium (pp. 241-246). Brighton: The Institute of Mining and Metalurgy.
Palmstrom (1996). Characterizing rock masses by the RMi for use in practical rock engineering, Part 2: Some practical applications of the Rock Mass index (RMi). Tunnelling and Undergroung Space Technology, 11(3): 287-303, doi: 10.1016/0886-7798(96)00028-4
Palmström (2009). Combining the RMR, Q, and RMi classification systems. Tunnelling and Underground Space Technology, 24(4): 491-492, doi: 10.1016/j.tust.2008.12.002.
Protodyakonov. (1907). Rock pressure on mine support (theory of mine support), pp. 23-45. Yekaterinoslav: Tipografiya Gubernskogo Zemstva
Ranasooriya and Nikraz (2009). Reliability of the linear correlation of Rock Mass Rating (RMR) and Tunnelling Quality Index (Q). Australian Geomechanics, 44(2): 47-54
Rutledge and Preston (1978). Experience with engineering classifications of rock. In International Tunnelling Symposium (pages A3.1-A3.7). Tokyo, Japan Tunnelling Association.
Sayeed and Khanna (2015). Empirical correlation between RMR and Q systems of rock mass classification derived from Lesser Himalayan and Central crystalline rocks. In International Conference on “Engineering Geology in New Millenium”. New Delhi: Journal Engineering of Geology.
Terzaghi (1946). Rock defects and loads on tunnel supports. In Proctor, R. V., White, T. L. (Eds.), Rock tunnelling with steel supports (pp. 17-99). Youngstown, Ohio: Commercial Shearing and Stamping Company.
Tzamos and Sofianos (2007). A correlation of four rock mass classification systems through their fabric indices. International Journal of Rock Mechanics and Mining Sciences, 44(4): 477-495, doi: 10.1016/j.ijrmms.2006.08.003.
Wickham et al. (1972). Support determinations based on geologic predictions. In North American rapid excavation and tunnelling conference (43-64). Chicago: Society of Mining Engineers, AIME.
Author Response
First of all, it must be said that, according to the journal’s template, references must be ordered as they appear in the paper, not in other order. Therefore, the first cite we find in the paper must be [1] and not [27] (line 37). Consequently, all the references must be reordered.
Thnak you for your remarks, the references and citations now follow the instructions given by Applied Sciences.
Secondly, I will extend the introduction of the rock mass classification. I see a gap between paragraph in lines 49-52 and subsequent paragraphs.
Once the necessity of the rock mass classification has been introduced in the paper, it should be mentioned that, various rock mass classifications appeared since the beginning of the 20th century. Some of those classifications were the system of Protodyaknov (1907), of Terzaghi (1946), the RQD (Deere et al. 1967), the RSR (Wickham et al. 1972), the RMR (Bieniawski, 1973), Q system (Barton et al. 1974), GSI (Hoek et al. 1995), RMi (Palmström, 1996), and some recent ones like the Rock Mass Fabric Indices (F) (Tzamos and Sofianos, 2007) and the Rock Mass Quality Index (Aydan et al. 2014). At this points, it should be mentioned that RMR and the Q system are the most widely employed methods (Palmströn, 2009; Ranassoriya and Nikraz 2009; Fernández-Gutierrez et al. 2017). However, other classifications like RMi and GSI are attracting more and more interest. Then, the RMR and the GSI can be described, continuing with existing text.
The authors modified the paper in accordance with the comments.
In line 53, I will remark that the RMR was developed by Bieniawski in 1973 [and its reference), to highlight that later, in 1989, the most used version emerged.
Thank you for your remark it is revised, now.
Similarly, in line 59 it is beneficial to introduce that Hoek et al. introduced the GSI in 1995. Thus, the time gap between both classifications can be better observed.
Thank you for your remark it is revised according to your comment.
Line 62-63. I think that this idea, that the GSI-based equations are sensitive to their respective input parameters is a universal idea that is verified for all the rock mass classifications. Therefore, it could be added, “As in other rock mass classifications, all the GSI-based equations are highly...”
The sentence was modified.
After line 64 or after line 66, I will include a very important idea: there are researchers in the literature about correlating rock mass classifications between them. Therefore, it could be included something like: “Although there are various rock mass classifications, no one of them have imposed above the other and normally, the employment of more than one classification is highly recommended, even by the authors of them (Bieniawski, 1989, Palmströng, 2009). Since the first correlation between RMR and the Q index, proposed by Bieniawski (1976), many correlations have been proposed from tunnel and mine projects all over the world (Rutledge and Preston, 1978; Moreno Tallón 1982, Sayeed and Khanna 2015). Moreover, when correlations are applied to the same rock lithology, better results are obtained (Castro-Fresno et al. 2010, Fernández-Gutierrez et al. 2017; Campos et al. 2020).
Thank you. The paper was corrected according to this suggestion.
Line 77. Fig 1 cannot be presented before it is commented in the text.
Thank you, it is revised.
Last but not least, for section 3 and 4, a better organization is needed. Available data must be better explained. I am really confused about the data that were analyzed. Data were obtained by observing in field. Then, what is GSIcalc? In my opinion, if authors have RMR and GSI data from different tunnel/mines, they can show the scatter plots and obtained correlations are presented. It is not necessary to calculate GSI from other proposed equations. It seems that authors try to verify others’ equations instead of developing their own correlations. A deep review of the procedures iof section 4 is needed.
The paper was corrected according to some part of this suggestion. Athough we suggest it is needed to verify our data with different equations and approaches.
Moreover, when presenting equations like in Figure 6, each of the references must be linked, i.e. its reference in square brackets must be given [xx]. Similarly, for Figure 7.
Thank you for the suggestions on formatting - it is now revised
New references
These new references were added to the mauscript as it was requested by the reviewer.
Round 2
Reviewer 2 Report
Dear authors, I am satisfied with the changes and explanations you have given. I wish you a lot of success in your future work!
Author Response
Thank you for your work and help - they were very useful
Reviewer 3 Report
The manuscript has been adequately improved. However, some modifications were not correctly made.
Figure 1 is referenced with reference [34], which is a document published by Barton. As the GSI was developed by Hoek, it is incorrect. Perhaps there is some confusion in the following references.
Table 1. For the column of the references, the names of the authors can be maintained, as it is easier to see their names. If the exact reference is wanted to be found, the number would indentify it in the reference list. Therefore, they should appear as:
For the first row: Hoek et al. (1995) [22] (but reference [22] is not from 1995).
For the second row: Cosar (2004) [42]
In the previous review I indicated that the references should be indicated in Figures 6 (Existing correlations between RMR89 and GSI listed in Table 1) and Figure 8 (The correltion between different a and b constants regarding different correlations between GSI-RMR89 values). In the legend, when indicating the authors of the correlations, their number of reference must be given: Osgoui and Ünal (2005) [xx]. The number in the square brackets is missing. Similarly for Figure 8, next to the names of the authors, the number of reference MUST be included.
Additionally, it must be said that the figures are not in order. This order can be seen in the paper:
Figure 1
Figure 6
Figure 3
Figure 4
Figure 5
Figure 6 (there are two Figure 6. That is why I included the title of the figure in the previous comment).
Figure 7
Figure 8.
Author Response
Thank you for your suggestions and help. Thank you for pointing out the errors.
Enclosed I am sending the new corrected version of the manuscript. We checked all the numbers (Eq., Fig, Table) and corrected them.